# A methodological assessment of randomization integrity in alteplase for acute ischemic stroke individual patient data meta-analyses

**Ravi Garg**[1]*, **Gabriel Torrealba-Acosta**[2], **Steffen Mickenautsch**[3], **Vance W. Berger**[4]

1 Department of Neurology, Division of Neurocritical Care, Loyola University Chicago Stritch School of Medicine, Maywood, Illinois, United States of America, 2 Department of Neurology, Duke University School of Medicine, Durham, North Carolina, United States of America, 3 Faculty of Dentistry, University of the Western Cape, Tygerberg/Cape Town, South Africa, 4 Biometry Research Group, National Cancer Institute, Rockville, Maryland, United States of America

☯ These authors contributed equally to this work.
* ragarg@luc.edu

## Abstract

### Objectives

Little is known about the integrity of randomization for randomized controlled trials (RCT) included in alteplase for stroke meta-analyses. If the RCTs were not properly randomized, the results could not be accepted at face value. The objective was to assess the integrity of randomization in individual patient data (IPD) meta-analyses supporting alteplase for acute ischemic stroke.

### Methods

We assessed randomization reporting, performed qualitative risk of bias assessments arising from the randomization process, and performed fixed effects meta-analyses of baseline variables for which zero heterogeneity is expected if all included RCTs have unbiased randomization. Fixed-effects meta-analyses of baseline age, weight, and National Institute of Health Stroke Scale (NIHSS) score were performed. If heterogeneity was present ($I^2 > 0\%$), trials were systematically removed, starting with the RCT with the largest t-statistic until the $I^2$ value was 0%.

### Results

The NINDS rt-PA Stroke Study had a high risk of bias, the ECASS-3 RCT had some concerns, and all other trials were graded as low risk according to the Cochrane Risk of Bias (ROB-2) tool. The NINDS rt-PA Stroke Study contributed to heterogeneity in age and weight meta-analyses, and the ECASS-3 RCT contributed to heterogeneity in the NIHSS score meta-analysis. Removal of suspect trials resulted in the expected $I^2$ value of 0%.

**Data availability statement:** Data Availability: The IST-3 RCT data set can be accessed here: https://github.com/ravigarg415/IST_3 while the NINDS rt-PA Stroke Study data set can be requested from the National Institute of Neurological Disorders and Stroke here: https://www.ninds.nih.gov/current-research/research-funded-ninds/clinical-research/archived-clinical-research-datasets.

**Funding:** The author(s) received no specific funding for this work.

**Competing interests:** The authors have declared that no competing interests exist.

## Conclusions:

The NINDS rt-PA Stroke Study and ECASS-3 trials contributed to heterogeneity in fixed effects meta-analyses of baseline variables while there should have been none. These RCTs are likely a source of selection bias in IPD meta-analyses due to suspect randomization.

## Introduction

Meta-analyses of randomized controlled trials (RCTs) are considered the highest level of evidence for clinical practice guideline (CPG) recommendations [1]. Meta-analyses are observational by nature and retain all the biases of component RCTs [2]. Component RCTs rely on randomization to control for selection bias and ensure the validity of statistical testing. Selection bias may be difficult to assess after randomization and depends on methodological reporting. Baseline imbalances following randomization may be a symptom of selection bias or a chance finding. The Cochrane Risk of Bias 2 (RoB 2) tool recommends assessing for baseline imbalances but emphasizes that "baseline differences that are compatible with chance do not lead to a risk of bias" [3]. A chance finding can only be assured, however, if both the random generation of the allocation sequence and the actual randomization procedure were unbiased [4]. For RCTs that employ highly restrictive randomization procedures, such as permuted block randomization, or that have poor methods of allocation sequence concealment, the assumption of unbiased randomization and baseline differences due to chance is a common misconception [4, 5]. Unlike missing outcome data, for which sensitivity analyses can be performed on summary data to ensure study results are robust, selection bias is assessed primarily by a qualitative audit of the randomization process. Adequate self-reporting of the randomization process, however, does not necessarily equate to effective randomization [4].

Hicks and colleagues have popularized an agnostic analysis to assess for bias introduced by component RCTs included in meta-analyses by inadequate allocation sequence concealment and other sources of biased randomization [6–12]. This method relies on the principle that there should be zero heterogeneity in a fixed-effects meta-analysis of baseline continuous variables from component RCTs.

A recent acute ischemic stroke CPG recommends alteplase therapy within 4.5 hours of symptom onset based on meta-analyses of RCTs [1]. Of the eight component RCTs used in recent individual patient data (IPD) meta-analyses, two have suffered from substantial baseline imbalances in important prognostic factors following randomization: the NINDS rt-PA Stroke Study and ECASS-3 [13, 14]. In the NINDS rt-PA Stroke Study, there were imbalances in baseline aspirin use; baseline stroke severity measured by the National Institute of Health Stroke Scale (NIHSS) score; stroke sub-type; and baseline computed tomography (CT) all favoring the alteplase arm [14]. In the ECASS-3 RCT, there was similarly an imbalance in the baseline NIHSS score and additionally, an imbalance in prior stroke status both favoring the alteplase arm [13]. Although multiple reanalyses have been published focusing on covariate adjustments, latent covariate imbalances cannot be accounted for if the randomization is not valid [15].

The possibility that component RCTs of alteplase treatment for acute ischemic stroke introduced bias arising from the randomization process into meta-analyses has not been assessed. Therefore, the purpose of this study was to assess for bias arising from the randomization process in component RCTs included in alteplase for ischemic stroke IPD meta-analyses.

## Methods

The publicly available NINDS rt-PA Stroke Study and IST-3 data sets were used for these analyses. The data was first accessed for research purposes on July 26th, 2022. The authors did not have access to information that could identify individual participants during or after data collection.

### Study design and population

Institutional review board or ethics approval was not obtained (exempt determination) based on guidance at the authors institutions. Randomized controlled trials used in IPD meta-analyses by the Stroke Thrombolysis Trialists' (STT) Collaborative Group were chosen for this analysis [16–21]. The STT Collaborative Group pooled IPD from nine RCTs for meta-analyses: NINDS A (N = 291); NINDS B (N = 333); ECASS I (N = 620); ECASS II (N = 800); ATLANTIS A (N = 142); ATLANTIS B (N = 613); ECASS III (N = 821); EPITHET (N = 101); and IST-3 (N = 3035) for a total of 6756 participants [22–29]. For this analysis, NINDS-A and NINDS-B were considered one trial as has been previously done given continuous, shared randomization between the two parts [15]. We excluded the ECASS 1 RCT, as has been previously done, to mimic treatment estimates used to support recommendations from clinical practice guidelines labeled "patients who would have met a 4.5 hour revised US label" by IPD meta-analysis authors [21]. The primary outcome of the meta-analysis was a modified Rankin Scale score of 0-1 at 3-6 months post-stroke. The Oxford Handicap Scale score (OHS) was treated as an equivalent to the mRS score.

### Data analysis

Data was abstracted from the published RCTs independently by two researchers (R.G. and G.T.A.). Data from the NINDS rt-PA Stroke Study and IST-3 RCT were abstracted from the publicly available datasets. Key components for randomization methods were assessed for all component RCTs. We assessed the risk of bias of RCTs included in the meta-analysis using the RoB 2 tool [3]. We compared our risk of bias assessments to those in the included meta-analyses.

The technique proposed by Hicks et al. was used for the primary analysis [6]. Three continuous variables were abstracted from component RCTs using a standardized form: age, NIHSS score, and weight. Age and NIHSS score were chosen due to their status as being the most important prognostic variables for ischemic stroke outcomes for which any mean difference between groups, irrespective of the p-value from null hypothesis significance testing (NHST), would be considered relevant in the assessment of post-randomization confounding [30]. Weight was chosen as a validation variable to ensure the integrity of the analysis due to consistent reporting in alteplase for stroke RCTs.

The t-statistic was calculated from an independent two-sample t-test for the difference in baseline continuous variables, and the studies were ranked, in descending order, by the absolute value of their t-statistic for each variable. Trials expressing baseline variables as medians were excluded. A fixed-effects meta-analysis for each variable was performed to measure heterogeneity by $I^2$ and trials were removed with subsequent iterations until the meta-analysis revealed no heterogeneity ($I^2 = 0\%$). To assess the assumptions of the procedure, it was repeated performing a random-effects meta-analysis (DerSimonian and Laird method) starting with the trial with the two smallest t-statistics and adding RCTs based on ascending order of their t-statistics until heterogeneity was observed. The point estimates and 95% confidence intervals for $I^2$ and $\tau^2$ were assessed at each iteration. The confidence intervals for $I^2$ and $\tau^2$ were determined using non-parametric bootstrapping and unequal tail probabilities [31].

All analyses were performed in R Studio version 2023.06.1 + 524. R code for the analysis can be provided for reasonable requests.

## Measures

To quantify the degree of selection bias, a pooled risk difference was determined with and without suspect trials. An mRS score of 0-2 was chosen to increase the precision of estimates.

## Results

### Randomization methods reporting and risk-*of*-bias arising *from the* randomization process

The randomization methods and risk-of-bias for the included RCTs were reported in Table 1.

Based on ROB-2, the NINDS rt-PA Stroke Study was rated as having a high risk of bias arising from the randomization process (S1 Table); while the ECASS-3 was rated as having some concerns (S4 Table). The remaining five RCTs [ATLANTIS (S2 Table); ECASS-2 (S3 Table); EPITHET (S5 Table); and IST-3 (S6 Table)] were rated as having a low risk of bias arising from the randomization process. In addition to being the only trials that were not graded as a low risk of bias, the NINDS rt-PA Stroke Study and ECASS-3 RCT were the only RCTs with baseline imbalances in important prognostic variables favoring the treatment arm (S1 Table; S4 Table). The risk of bias arising from the randomization process was not reported in the five IPD meta-analyses published.

### Heterogeneity *in* baseline variables

The differences in independent means of age, weight, and NIHSS score; and their associated t-statistics were reported in S7 Table. The EPITHET RCT only reported age as a mean ± SD while the NIHSS was reported as a median and weight was unreported [22]. The ECASS-2

**Table 1. Randomization reporting and risk of bias for included randomized trials.**

| Trial | Randomization Method | Allocation Sequence Concealment | Risk of Bias Arising from the Randomization Process |
|---|---|---|---|
| **NINDS rt-PA Stroke Study** | Stratified block randomization. Block sizes varied. | Non-numerically ordered drug pre-packs. Patient ID matched to randomization schedules kept at site of randomization. | High |
| **ECASS-2** | Stratified block randomization. Fixed block sizes of four. | Unreported, but noted to use "sequential patient numbers." | Low |
| **ATLANTIS**-A | Stratified block randomization. Block size unreported. | Interactive voice system | Low |
| **ATLANTIS-B** | Stratified block randomization. Block size unreported. | Interactive voice system | Low |
| **ECASS-3** | Permuted block randomization. Fixed blocks sizes of four. | Interactive voice system | Some Concerns |
| **EPITHET** | Permuted block randomization. Fixed block sizes of four. | Sequentially numbered drug pre-packs. | Low |
| **IST-3** | Minimisation. | Web-based or telephone system. | Low |

RCT did not report continuous baseline characteristics as means ± standard deviations or report interquartile ranges associated with medians and was excluded from the analysis [25]. The NINDS rt-PA Stroke Study had the highest t-statistic for age (2.19) and weight (2.69). The ECASS-3 RCT had the highest t-statistic for NIHSS (2.24) (S7 Table).

A fixed-effects meta-analysis for age revealed the NINDS rt-PA Stroke Study was the only study for which the 95% confidence interval did not include the null. Heterogeneity was present ($I^2 = 15\%$, 95% CI 0-50%) [Fig 1].

Exclusion of the NINDS rt-PA Stroke Study was required to achieve no heterogeneity ($I^2 = 0\%$, 95% CI 0-0%).

A fixed-effects meta-analysis for weight revealed the NINDS rt-PA Stroke Study was the only study for which the 95% confidence interval did not include the null. Heterogeneity was present ($I^2 = 45\%$, 95% CI 0-68%) [Fig 2]. Exclusion of the NINDS rt-PA Stroke Study was required to achieve no baseline heterogeneity ($I^2 = 0\%$, 95% CI 0-0%).

A fixed-effects meta-analysis for NIHSS revealed the ECASS 3 RCT was the only study for which the 95% confidence interval did not include the null. Heterogeneity was present ($I^2 = 31\%$, 95% CI 0-55%) [Fig 3]. Exclusion of the ECASS 3 RCT was required to achieve no baseline heterogeneity ($I^2 = 0\%$, 95% CI 0-0%). The procedure for all three variables was validated in reverse (S1–S3 Figs). The results were robust to changes in sample size and were identical using $\tau^2$.

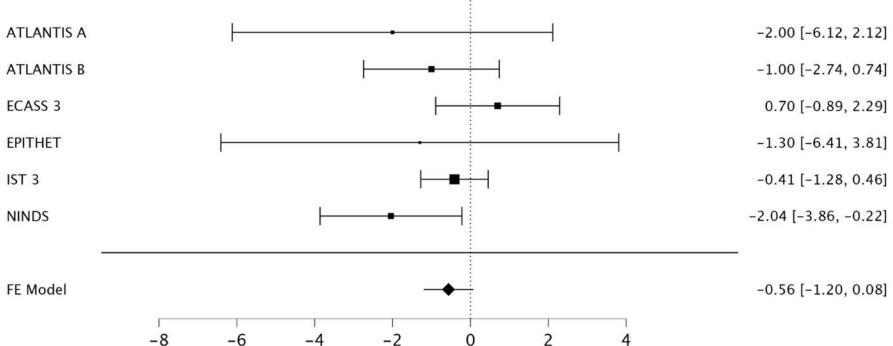

**Fig 1. Forest plot of mean differences for baseline age for included trials.** The NINDS rt-PA Stroke Study (NINDS) is the only included study for which the 95% confidence interval did not include the null.

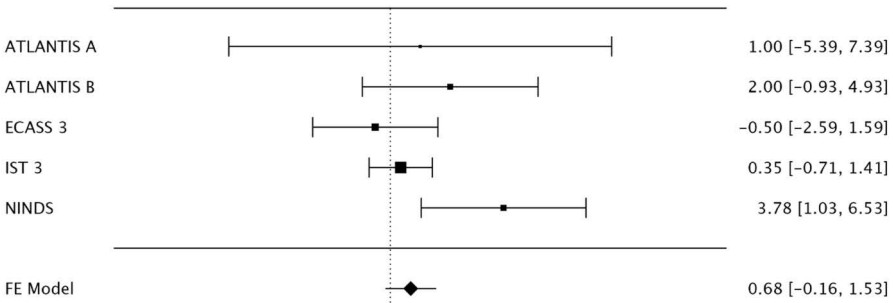

**Fig 2. Forest plot of mean differences for baseline weight for included trials.** The NINDS rt-PA Stroke Study was the only included study for which the 95% confidence interval did not include the null.

## Quantification of selection bias

The pooled absolute risk difference, using the DerSimonian-Laird method, for all included trials was 3% (95% CI, -1% - 8%) (Fig 4). After the exclusion of the NINDS rt-PA Stroke Study and ECASS-3, the pooled absolute risk difference was reduced to 1% (95% CI, -4% - 6%) (Fig 5). The I² estimate including all studies was 58% compared to 50% after removal of the NINDS rt-PA Stroke Study and ECASS-3.

## Discussion

Qualitative risk of bias assessments arising from the randomization process, based on ROB 2, graded the NINDS rt-PA Stroke Study as a high risk of bias while the ECASS-3 RCT had some concerns. Both trials inadequately reported random sequence generation and randomization implementation, and both trials contributed to heterogeneity in the fixed-effects meta-analyses of baseline variables. Taken together, these trials are a source of systematic error in

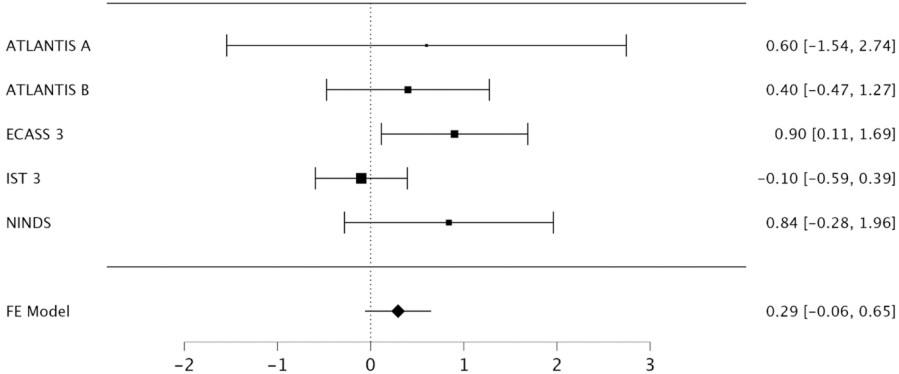

**Fig 3. Forest plot of mean difference for baseline NIHSS for included trials.** The ECASS-3 RCT was the only included study for which the 95% confidence interval did not include the null.

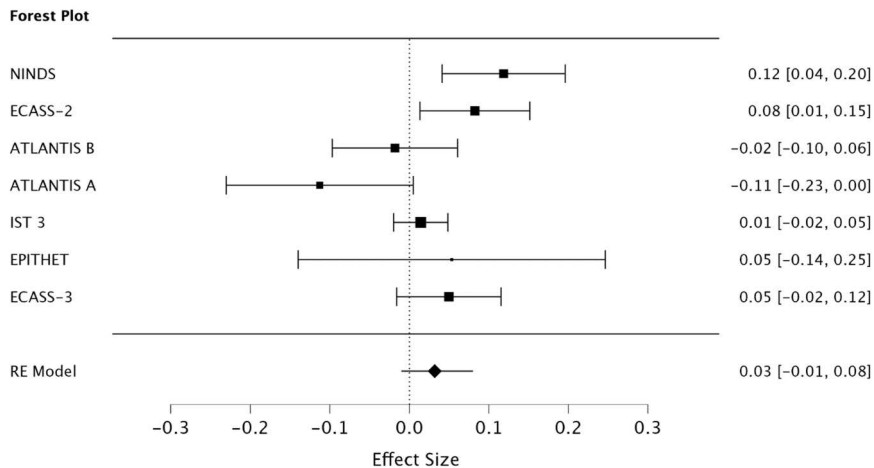

**Fig 4. Pooled absolute risk difference for all included trials.**

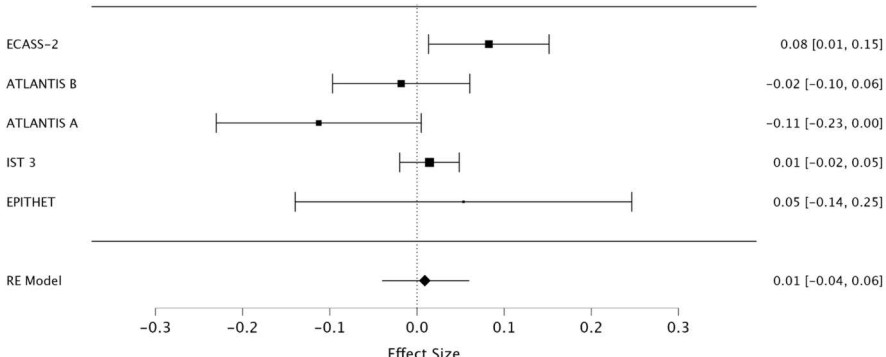

**Fig 5. The pooled effect of treatment with and without the NINDS rt-PA Stroke Study and ECASS-3 RCT.**

meta-analyses. We estimated that 2% of the average treatment effect (ATE) can be attributed to selection bias.

Our results found congruence between the qualitative risk of bias assessments and the evaluation of heterogeneity in the fixed-effects meta-analyses of baseline variables such that the two trials that were not graded a low-risk of bias (NINDS rt-PA Stroke Study and ECASS-3 RCT), also contributed to heterogeneity in the meta-analyses. Other authors, however, have reported discrepant results between the qualitative risk of bias assessments and the evaluation of heterogeneity in fixed-effects meta-analyses of baseline variables. For example, in a meta-analysis of RCTs of transversus abdominis plane block efficacy in hysterectomy, six RCTs were graded with a low risk of bias arising from the randomization process, and one was rated as having some concerns. Despite the favorable qualitative risk of bias assessments, meta-analyses of two baseline variables identified heterogeneity, while there should have been none if randomization was unbiased [6].

There are other limitations to full reliance on the qualitative risk of bias assessments. The ROB 2 tool is dependent on the self-reporting of RCTs. For instance, in a meta-analysis on the effect of vitamin K supplementation on bone mineral density and fractures, most trials had an unclear risk of bias in the random sequence generation or allocation sequence concealment domains [10]. A fixed-effects meta-analysis of baseline age revealed no heterogeneity providing reassurance about the integrity of randomization in these individual studies. These examples support previous recommendations on more detailed reporting in RCTs to control for selection bias (Table 2) [32].

The NINDS rt-PA Stroke Study and ECASS-3 RCT have similarities that are worth noting. Both RCTs had baseline imbalances in important prognostic variables (S1 and S4 Tables) and were also the only trials that contributed to heterogeneity in meta-analyses of baseline variables. The use of NHST to assess for differences at baseline is controversial [33]. A major argument against the routine use of NHST is that unbiased randomization produces an expected distribution of $p$ values that does not necessarily apply to a single hypothesis test. Additionally, this distribution may be affected by the correlation between baseline characteristics and the number of hypothesis tests performed. Conversely, the premise of these arguments is that randomization, and not just random sequence generation, is unbiased which may be a foregone conclusion. Our results support the use of NHST at baseline and are consistent with other RCTs for which substantial baseline imbalances, coupled with flawed randomization methods, raised suspicion of selection bias [4].

Both RCTs employed permuted block randomization for which the potential for third-order selection bias has been thoroughly described [5]. Permuted blocks facilitate

predictability in future allocations and may lead to more favorable patient selection in the experimental arm, and this has been referred to as the convergent strategy of guessing without certainty [34]. In the NINDS rt-PA Stroke Study, varying block sizes were used. Contrary to popular belief, varying the block size does not lessen the chance of randomization subversion [34, 35]. Rather, this strategy may be as problematic as using fixed block sizes. In a review of 179 open-label RCTs employing the same methodology chosen for the current report, the authors found baseline heterogeneity for age in all trials implementing a block randomization scheme, trials implementing a fixed block size of 4, and trials implementing varied block sizes including a block size of 2 [36]. In the ECASS-3 RCT, a fixed block size of four was used.

The principal issue with varied block sizes, which are smaller on average, and a fixed block size of four is the same: high predictability of future allocations. To illustrate, if the first allocation is identified as alteplase, the second allocation is twice as likely to be placebo as alteplase using a block size of 4. If a record of allocations is kept the last allocation is 100% predictable in 1/3$^{rd}$ of block sequence permutations [36]. Even if the allocation sequence is strictly concealed, the potential for accidental unblinding in alteplase trials is high by intracerebral hemorrhage, systemic bleeding, or angioedema. Additionally, neither the NINDS rt-PA Stroke Study nor the ECASS-3 RCT reported how the frothing reaction that occurs with alteplase reconstitution was mimicked in the placebo arm as is explicitly reported in more modern alteplase RCTs [37].

The EPITHET and ATLANTIS RCTs also employed blocked randomization. Still, they did not contribute to baseline heterogeneity in the age meta-analysis suggesting other elements of susceptible randomization may be required to precipitate randomization subversion. For example, randomization was decentralized in the NINDS rt-PA Stroke Study. Other elements that precipitate randomization subversion, such as the motivation of investigators or conflicts of interest, are unlikely to be fully substantiated post hoc.

Although permuted blocks facilitate the convergent strategy of guessing and may introduce third-order selection bias, the ability to directly subvert randomization is also possible; and this has been suspected in the NINDS rt-PA Stroke Study [14]. In this trial, sealed envelopes with the treatment assignments were attached to study pre-packs for purposes of emergent unblinding for adverse events. At the end of the trial, sixteen envelopes were opened for unblinding of which eight did not have listed safety reasons. Comparatively, only five envelopes were opened for unblinding in the ECASS-2 RCT which had a larger sample size. The NINDS rt-PA Stroke trial data revealed deviations from the expected 1:1 allocation ratio in

**Table 2. Recommendations for improved randomization reporting in randomized controlled trials. Adapted from Berger and Cristophi [32].**

| Concern | Reporting Recommendation |
| --- | --- |
| Allocation Discretion | Planned allocation proportions and numbers of screened/randomized patients by assigned group. |
| Deferred Enrollment | Indicate if any participants were screened multiple times or confirm that no duplicates occurred. |
| Allocation Concealment | Explain how future group assignments were kept secret. |
| Predicted Allocations | Provide details on randomization methods, including any restrictions like block sizes, and describe how group assignments were masked and any instances where assignments were revealed. |
| Baseline Imbalances | Assess and report whether participant characteristics, especially when prognostic, were evenly distributed between groups. |
| Selection Bias | Plot key variables and outcomes against the likelihood of being assigned to each group (reverse propensity score) and note any errors in stratification. |

most randomization strata; violation of a pre-specified randomization rule at 3 centers; and multiple baseline imbalances in prognostic variables favoring the alteplase arm when evaluated on exclusion criteria (S8 Table) [38]. A similarly detailed audit of the randomization process of other included trials, including the ECASS-3 RCT, cannot be performed due to a lack of publicly available trial data, study protocols, or product licensing applications.

Our results have important implications for meta-analyses and CPG recommendations regarding alteplase for acute ischemic stroke. First, we found that meta-analyses, including the NINDS rt-PA Stroke Study and ECASS-3, overestimated the ATE by 2%. Importantly, covariate adjustments do not adequately control this bias, as flawed randomization might lead to latent covariate imbalances. To perform a meta-analysis without removing suspect trials, the reverse propensity score (RPS) could be used to correct for selection bias [39]. The RPS is the conditional probability of a patient being assigned a specific treatment given prior allocations under block randomization. The randomization schedules are required to calculate the RPS, however, and are not publicly available.

As meta-analyses are widely considered the pinnacle of the evidence-based medicine pyramid, our findings have important implications for CPG recommendations. Currently, CPGs do not acknowledge the presence of selection bias in the NINDS rt-PA Stroke Study and ECASS-3 [1,40]. Aligned with the best methodological practices for CPG development, we believe the strength of the recommendation, currently "strong", should be reconsidered based on limitations in the quality of the evidence [41].

Despite the inherent risk of selection bias, permuted block randomization remains a common randomization procedure in stroke trials and may be a more pervasive problem than appreciated. For example, a recent thrombolytic trial employing stratified block randomization resulted in unequal group allocations by the stratifying factor and ten baseline factors favoring the thrombolytic arm concerning for selection bias [42, 43]. Therefore, methods that are superior to permuted block randomization, such as the asymptotic maximal procedure, simple randomization, and minimization, should be used. Bayesian adaptive randomization designs have also recently been successfully employed in stroke clinical trials [44]. Alternatively, if permuted block randomization is performed, the integrity of the process should be verified using the RPS.

## Limitations

The current report has limitations that are worth noting. The IPD meta-analyses include the ECASS-2 RCT which was excluded in the current report due to a lack of reporting of the chosen continuous variables. In ECASS-2, the median age and NIHSS were reported without their associated interquartile ranges. We believe this deviates from the best practices of baseline reporting. As previously noted, only trials that were not graded with a low risk of bias contributed to heterogeneity while ECASS-2 was graded with a low risk of bias. Given the heterogeneity contributions of the NINDS rt-PA Stroke Study and ECASS-3 RCT, it is unlikely that the exclusion of ECASS-2 meaningfully changes the conclusions as we also validated the procedure in reverse.

We found discrepancies in the trials that contributed to heterogeneity in meta-analyses: the NINDS rt-PA Stroke Study contributed to heterogeneity in age and weight meta-analyses while the ECASS-3 RCT only contributed to heterogeneity in the NIHSS score meta-analysis. It is notable in the meta-analysis of baseline NIHSS, that the 95% confidence interval begins to increase only after the addition of the NINDS rt-PA Stroke Study when the procedure was conducted in reverse, although the point estimate remained 0 (S3 Fig). Importantly, the chosen variables may not reflect the baseline characteristic that randomization was subverted on as this may be an unknown or categorical covariate. Rather,

there may be an association between the variable for which heterogeneity was detected and the baseline characteristic(s) that randomization was subverted on. Alternatively, the discrepancies in heterogeneity may simply be a chance finding and this has been previously described using the current methodology in a systematic review of spinal manipulative therapy for chronic low back pain [7]. We agree with the original authors of the current method that any observed baseline heterogeneity is a cause for concern. As such, the reported results are best served in congruence, as opposed to isolation, with trial-level data. In the case of the NINDS rt-PA Stroke Study, the reported results are consistent with a risk of selection bias assessment using trial-level data [14]. A similar trial-level assessment of the ECASS-3 RCT could confirm or deny these findings.

We were unable to replicate multiple subgroup analyses published in IPD meta-analyses. Any subgroup, however, would retain systematic error found in the pooled estimate of all trials. The current report also only addressed bias arising from the randomization process. Other limitations of the IPD meta-analyses such as unblinding in the IST-3 RCT; handling of missing outcomes; and inclusion of RCTs with small sample sizes were not addressed in the current report and should also be considered [45–47].

Due to the lack of diversity in randomization methods used in the included trials, we could not substantiate if certain randomization procedures inherently impact the detection of baseline heterogeneity in the current method. Five out of six included trials used permuted block randomization, and one used minimization. Due to the predictability of future allocations, we believe this randomization method has a specifically higher risk of selection bias than others such as minimization with a random factor as used in the IST-3 RCT.

Finally, as previously noted, we did not have access to the randomization schedules and were unable to calculate an RPS to definitively detect and correct for selection bias; or assess for over-stratification.

A strength of the current report is the use of agnostic, validated analysis using summary data to assess randomization. The procedure has been reported in meta-analyses of tranexamic acid for post-partum hemorrhage; vitamin K supplementation on bone mineral density and fractures; transversus abdominis plane block efficacy in hysterectomy; interventions for the management of primary frozen shoulder; and atypical antipsychotics in dementia suggesting strong external validation [8–12]. The procedure was robust to assumptions, including changes in sample size, consistent with a recent simulation [48]. An important future area of meta-epidemiological research is determining, using the current method, whether certain randomization procedures portend a higher risk of selection bias than others.

## Conclusions

Among RCTs included in IPD meta-analyses of alteplase for acute ischemic stroke, the NINDS rt-PA Stroke Study had a high risk of bias arising from the randomization process while the ECASS-3 RCT had some concerns. Both trials inadequately reported randomization and contributed to heterogeneity in fixed effects meta-analyses of baseline variables suggesting biased randomization. Results from these RCTs, and meta-analyses that include these RCTS, should be accepted cautiously.

## Supporting Information

**S1 Table: Risk of Bias Arising from the Randomization Process in the NINDS rt-PA Stroke Study.**
(DOCX)

**S2 Table: Risk of Bias Arising from the Randomization Process in the ATLANTIS-A and ATLANTIS-B Randomized Clinical Trials.**
(DOCX)

**S3 Table: Risk of Bias Arising from the Randomization Process in the ECASS-2 Randomized Clinical Trial.**
(DOCX)

**S4 Table: Risk of Bias Arising from the Randomization Process in the ECASS-3 trial.**
Abbreviations: NIHSS, National Institute of Health Stroke Scale.
(DOCX)

**S5 Table: Risk of Bias Arising from the Randomization Process in the EPITHET trial.**
(DOCX)

**S6 Table: Risk of Bias Arising from the Randomization Process in the IST-3 trial.**
(DOCX)

**S7 Table: Mean differences in age, weight, and National Institute of Health Stroke Scale scores; and the associated t-statistic for all included trials.**
(DOCX)

**S8 Table. Overview of selection bias in the NINDS rt-PA Stroke Study.**
(DOCX)

**S1 Fig. The $I^2$ point estimate (orange dot) and associated 95% confidence intervals for the age meta-analysis**
(TIF)

**S2 Fig. The $I^2$ point estimate (orange dot) and associated 95% confidence intervals for the weight meta-analysis.**
(TIF)

**S3 Fig. The $I^2$ point estimate (orange dot) and associated 95% confidence intervals for the NIHSS meta-analysis.**
(TIF)

## Acknowledgments

We thank the patients who participated in the included stroke trials.

## Author contributions

**Conceptualization:** Ravi Garg, Steffen Mickenautsch.

**Data curation:** Ravi Garg, Gabriel Torrealba-Acosta.

**Formal analysis:** Ravi Garg, Gabriel Torrealba-Acosta.

**Methodology:** Ravi Garg, Gabriel Torrealba-Acosta, Steffen Mickenautsch, Vance W. Berger.

**Supervision:** Steffen Mickenautsch, Vance W. Berger.

**Writing – original draft:** Ravi Garg, Gabriel Torrealba-Acosta, Steffen Mickenautsch, Vance W. Berger.

**Writing – review & editing:** Ravi Garg, Gabriel Torrealba-Acosta, Steffen Mickenautsch, Vance W. Berger.

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
