## [Decision Letter · Decision Letter 0]

9 Dec 2024

PONE-D-24-20124A methodological assessment of randomization integrity in alteplase for acute ischemic stroke individual patient data meta-analysesPLOS ONE

Dear Dr. Torrealba-Acosta,

Thank you for submitting your manuscript to PLOS ONE. After careful consideration, we feel that it has merit but does not fully meet PLOS ONE’s publication criteria as it currently stands. Therefore, we invite you to submit a revised version of the manuscript that addresses the points raised during the review process.

We look forward to receiving your revised manuscript.

Kind regards,

Cem Bilgin

Academic Editor

PLOS ONE

Reviewers' comments:

Reviewer's Responses to Questions

**Comments to the Author**

1. Is the manuscript technically sound, and do the data support the conclusions?

Reviewer #1: Yes

Reviewer #2: Yes

Reviewer #3: Yes

2. Has the statistical analysis been performed appropriately and rigorously? 

Reviewer #1: Yes

Reviewer #2: Yes

Reviewer #3: Yes

3. Have the authors made all data underlying the findings in their manuscript fully available?

Reviewer #1: Yes

Reviewer #2: Yes

Reviewer #3: Yes

4. Is the manuscript presented in an intelligible fashion and written in standard English?

Reviewer #1: Yes

Reviewer #2: Yes

Reviewer #3: Yes

5. Review Comments to the Author

Reviewer #1: This manuscript provides a valuable critical assessment of randomization integrity in pivotal alteplase trials for ischemic stroke, identifying concerning baseline imbalances in two key studies. While the methodology is sound, further clarity on the broader clinical implications and potential methodological enhancements for future trials would strengthen the study's impact. By addressing these limitations, the authors could contribute to more robust guidelines and better-supported conclusions in stroke treatment research.

1. The manuscript highlights significant baseline imbalances in the NINDS and ECASS-3 studies, suggesting potential selection bias. Could you discuss if these imbalances could be mitigated or accounted for in future meta-analyses, perhaps by employing statistical adjustment techniques? How might this affect the robustness of the meta-analytic findings?

2. Your findings raise questions about the integrity of widely cited alteplase trials. Could you elaborate on how your results might impact current clinical guidelines for ischemic stroke treatment? Are there implications for ongoing or future clinical trials on stroke treatment?

3. The study depended on publicly available datasets and could not perform a detailed review of the randomization protocols for all included trials. Could you clarify if future research with access to detailed randomization logs and protocols might yield different results? Additionally, are there specific data points that, if available, would strengthen the conclusions drawn?

4. Given the variability in randomization approaches across the RCTs analyzed, could you elaborate on how these differences might inherently impact baseline heterogeneity? It would be beneficial to explain if specific randomization methods were more prone to bias, as this could guide future research designs.

Reviewer #2: 1- Detailed analyses are made and supported by graphs.

2- The number of references is sufficient.

3- Although there are sometimes complex narratives, the sentences are made meaningful and understandable by collecting the sentences afterwards.

4- Although the study investigates the methods used in alteplase meta-analyses for stroke, it provides seminal information and recommendations for the evaluation of many studies.

5- A very good meta-analysis study with statistical data. It is a study that will be taken as an example with its many perspectives and applications and will contribute to the literature. Thank you to the authors. I recommend it to be accepted for publication in your journal.

Reviewer #3: This is a well written methodologic overview of methodological assessment of randomization integrity.

The authors note that based on ROB-2, the NINDS rt-PA Stroke Study was rated as having a high risk of bias arising from the randomization process (S1 Table ); while the ECASS-3 was rated as having some concerns (S4 Table). What exactly was the signal to point to actual risk? Was it insufficient stratification factors up front leading to imbalanced overall? The last column of Table S1 and S4 are after the fact to show imbalance perhaps. Is there any other indicator available?

Although the block sizes on the Figures in the supplement are an indication of the relative study size, it would be helpful to have more descriptive information about the actual individual sample sizes or other relevant information.

6. PLOS authors have the option to publish the peer review history of their article (what does this mean? ). If published, this will include your full peer review and any attached files.

**Do you want your identity to be public for this peer review?** For information about this choice, including consent withdrawal, please see our Privacy Policy .

Reviewer #1: No

Reviewer #2: **Yes: ** Mehmet Fatih INECIKLI

Reviewer #3: No

---

## [Author Response · Author response to Decision Letter 1]

17 Jan 2025

Manuscript Title: A methodological assessment of randomization integrity in alteplase for acute ischemic stroke individual patient data meta-analyses

Journal: PLOS ONE

Manuscript ID: PONE-D-24-20124

Date: 12/13/2024

Dear Dr. Cem Bilgin,

We appreciate the constructive feedback and insightful comments provided by the reviewers. We have carefully considered each point and made the necessary revisions to enhance the clarity and scientific rigor of our manuscript. There were no direct responses to Reviewer 2 based on their comments. The “Track Changes” option was employed in Microsoft Word and all new text was highlighted in yellow.

Below, we provide detailed responses to each comment, including specific line numbers where changes have been implemented in the revised manuscript. For clarity, we have structured our responses in a tabular format.

We trust that these revisions address the reviewers' concerns. We are grateful for the opportunity to improve our manuscript and look forward to your feedback. Please do not hesitate to contact us if further clarification is needed.

Sincerely,

Ravi Garg, MD

Gabriel Torrealba-Acosta, MD

Steffen Mickenautsch,BDS, Ph.D

Vance W. Berger, Ph.D

Reviewer 1 Comment Response

The manuscript highlights significant baseline imbalances in the NINDS and ECASS-3 studies, suggesting potential selection bias. Could you discuss if these imbalances could be mitigated or accounted for in future meta-analyses, perhaps by employing statistical adjustment techniques? How might this affect the robustness of the meta-analytic findings?

We appreciate your comments which improved the quality of the manuscript. Please see the added text in lines 770 and 795; 832-840; and 856-858.

While known covariate imbalances can be adjusted for, latent or unknown covariates imbalances cannot be adjusted for using common methods. Rather, a reverse propensity score can be used to determine and correct for selection bias.

We estimated the average treatment effect for the meta-analyses that include the suspect trials will be overestimated by 2%.

Your findings raise questions about the integrity of widely cited alteplase trials. Could you elaborate on how your results might impact current clinical guidelines for ischemic stroke treatment? Are there implications for ongoing or future clinical trials on stroke treatment?

Currently, the most cited clinical practice guidelines do not acknowledge the presence of selection bias in the NINDS rt-PA Stroke Study or ECASS-3. Best methodological practices for guideline development recommend downgrading recommendations for evidentiary quality concerns. Therefore, we advocate for guidelines to reassess their recommendations.

Despite the inherent risk of selection bias under a permuted block randomization procedure, this remains one of the most common procedures overall. We have provided evidence of selection bias from a very recently published thrombolytic trial to support our concerns. Other randomization procedures such as simple randomization, urn randomization, the asymptotic maximal procedure or minimization should be used.

The study depended on publicly available datasets and could not perform a detailed review of the randomization protocols for all included trials. Could you clarify if future research with access to detailed randomization logs and protocols might yield different results? Additionally, are there specific data points that, if available, would strengthen the conclusions drawn?

We have added a table (Table 2) that outlines necessary reporting to mitigate concerns for selection bias. The most necessary, single piece of data necessary to assess for selection bias, under permuted block randomization, is the randomization schedule.

Given the variability in randomization approaches across the RCTs analyzed, could you elaborate on how these differences might inherently impact baseline heterogeneity? It would be beneficial to explain if specific randomization methods were more prone to bias, as this could guide future research designs.

We would like to clarify that only one trial of the selected randomized trials did not use permuted block randomization while all other did. As such, assessing if different randomization procedures have a greater chance of contributing to baseline heterogeneity was not possible in our included studies. However, we agree that a larger meta-epidemiological study is worthwhile given the frequency that this method has been employed.

Reviewer 3 Comments Response

What exactly was the signal to point to actual risk?

Was it insufficient stratification factors up front leading to imbalanced overall? The last column of Table S1 and S4 are after the fact to show imbalance perhaps. Is there any other indicator available?

We appreciate your comments which improved the quality of our manuscript. Please see the new supplementary table 8 (S8 Table). Please also see the revised S1 and S2 Figures.

Historically, covariate imbalances is what raised concerns of selection bias in the NINDS rt-PA Stroke Study and ECASS-3. To fully answer your question, please see the newly added STable 8 and citation 14. There were both risk factors, a plausible mechanism and evidence from trial data that made selection bias manifest. We cannot, in the same way, substantiate this claim for ECASS-3 due to lack of publicly available study protocols, trial data and regulatory documentation. You are correct that imbalances are indeed after the fact (outcome of selection bias), although they are often the first clue to the presence of selection bias.

While intuitively stratification might reduce covariate imbalances, stratification factors cannot avert selection bias which is systematic error in group allocation. In certain cases, such as stratification by site of recruitment, predictability in next group allocation may be higher and worsen the chance of selection bias.

Although the block sizes on the Figures in the supplement are an indication of the relative study size, it would be helpful to have more descriptive information about the actual individual sample sizes or other relevant information.

See revised S1-S3 figures.

---

## [Editor Report · Decision Letter 1]

4 Feb 2025

A methodological assessment of randomization integrity in alteplase for acute ischemic stroke individual patient data meta-analyses

PONE-D-24-20124R1

Dear Dr. Torrealba-Acosta,

We’re pleased to inform you that your manuscript has been judged scientifically suitable for publication and will be formally accepted for publication once it meets all outstanding technical requirements.

Kind regards,

Cem Bilgin

Academic Editor

PLOS ONE
---

## [Editor Report · Acceptance letter]

PONE-D-24-20124R1

PLOS ONE

Dear Dr. Torrealba-Acosta,

I'm pleased to inform you that your manuscript has been deemed suitable for publication in PLOS ONE. Congratulations! Your manuscript is now being handed over to our production team.

Kind regards,

on behalf of

Dr. Cem Bilgin

Academic Editor

PLOS ONE